# Plant-Derived Nutraceuticals Involved in Body Weight Control by Modulating Gene Expression

**DOI:** 10.3390/plants12122273

**Published:** 2023-06-11

**Authors:** Maria Vrânceanu, Simona-Codruţa Hegheş, Anamaria Cozma-Petruţ, Roxana Banc, Carmina Mariana Stroia, Viorica Raischi, Doina Miere, Daniela-Saveta Popa, Lorena Filip

**Affiliations:** 1Department of Toxicology, “Iuliu Haţieganu” University of Medicine and Pharmacy, 6 Pasteur Street, 400349 Cluj-Napoca, Romania; marievranceanu@gmail.com (M.V.); dpopa@umfcluj.ro (D.-S.P.); 2Department of Drug Analysis, “Iuliu Haţieganu” University of Medicine and Pharmacy, 6 Pasteur Street, 400349 Cluj-Napoca, Romania; 3Department of Bromatology, Hygiene, Nutrition, “Iuliu Haţieganu” University of Medicine and Pharmacy, 6 Pasteur Street, 400349 Cluj-Napoca, Romania; anamaria.cozma@umfcluj.ro (A.C.-P.); roxana.banc@umfcluj.ro (R.B.); dmiere@umfcluj.ro (D.M.); lfilip@umfcluj.ro (L.F.); 4Department of Pharmacy, Oradea University, 1 Universităţii Street, 410087 Oradea, Romania; carmina.marian@yahoo.com; 5Laboratory of Physiology of Stress, Adaptation and General Sanocreatology, Institute of Physiology and Sanocreatology, 1 Academiei Street, 2028 Chișinău, Moldova; vioricalana@gmail.com

**Keywords:** obesity, nutraceuticals, epigenetics, gene expression, weight loss

## Abstract

Obesity is the most prevalent health problem in the Western world, with pathological body weight gain associated with numerous co-morbidities that can be the main cause of death. There are several factors that can contribute to the development of obesity, such as diet, sedentary lifestyle, and genetic make-up. Genetic predispositions play an important role in obesity, but genetic variations alone cannot fully explain the explosion of obesity, which is why studies have turned to epigenetics. The latest scientific evidence suggests that both genetics and environmental factors contribute to the rise in obesity. Certain variables, such as diet and exercise, have the ability to alter gene expression without affecting the DNA sequence, a phenomenon known as epigenetics. Epigenetic changes are reversible, and reversibility makes these changes attractive targets for therapeutic interventions. While anti-obesity drugs have been proposed to this end in recent decades, their numerous side effects make them not very attractive. On the other hand, the use of nutraceuticals for weight loss is increasing, and studies have shown that some of these products, such as resveratrol, curcumin, epigallocatechin-3-gallate, ginger, capsaicin, and caffeine, can alter gene expression, restoring the normal epigenetic profile and aiding weight loss.

## 1. Introduction

Despite the degree of malnutrition existing on the planet, according to the World Health Organization (WHO), obesity is one of the main public health problems in the world. In fact, we are facing a real global epidemic, which is spreading in many countries and which, in the absence of immediate action, could cause very serious health problems in the coming years. There are 375 million women and 266 million men who are overweight or obese in the world, and the US is at the top of obesity rankings. Among the most developed countries, Japan is the one in which the inhabitants have the lowest body mass index. In Europe, the most in line are Swiss women and Bosnian men. Globally, 2.3% of men and 5% of women are considered severely obese, that is, with a BMI above 35. Continuing at the current rate, 18% of men and 21% of women will suffer from severe obesity by 2025 [1].

Obesity is a condition characterized by excessive body weight due to the accumulation of adipose tissue, which develops due to the interaction of various factors, including genetic, endocrine-metabolic, and environmental factors. Therefore, it is a very common chronic condition that can negatively affect the state of health because it increases the risk of developing other diseases and worsens the person’s quality of life [2].

Indeed, obesity has a wide range of health impacts. Obese individuals are in fact at greater risk for the development of various disorders, including metabolic diseases such as diabetes and dyslipidemia (high levels of cholesterol and triglycerides), cardiovascular diseases such as stroke and heart attack, respiratory diseases, joint problems, gynecological disorders (menstrual irregularities, polycystic ovary syndrome, pregnancy complications), infertility, sexual disorders (impotence), predisposition to the development of diseases of the digestive system (e.g., gastroesophageal reflux, gallbladder stones), and mood disorders (e.g., depression) [3]. Finally, it should be remembered that the presence of obesity increases the risk of developing certain tumors, such as endometrial cancer (a type of uterine cancer), colorectal, gallbladder, and breast cancer [4]. This is why it is essential to understand basic cellular and molecular mechanisms in order to identify new therapeutic targets against obesity.

On a psychological level, obesity can completely turn a person’s life upside down; those who are obese are often isolated and subjected to social stigmatization, which makes any type of sociability difficult. In particular, overweight children tend to develop a difficult relationship with their bodies and with their peers, resulting in isolation, which often translates into further sedentary habits [5].

The recent “Human Obesity Gene Map” [6] lists 11 single gene mutations, 50 related loci to the relevant Mendelian syndromes in obesity, 244 transgenic or “knockout” animal models, and 127 gene candidates, of which just under 20% are replicates in more than five studies; a total of 253 “quantitative trait loci” for different phenotypes related to obesity are related to 61 genomes, and of these only about 20% are supported by more than one study [7]. Even if GWAS studies have identified hundreds of loci associated with BMI, this correlation can explain only 3–5% of BMI variance in the population [8].

The inability to correlate obesity to specific genes in a marked way has prompted research towards epigenetic studies. In fact, the genes not being sufficient to explain what happens in the phenotype, other forms of variation such as epigenetic markers must be taken into consideration. It is already known that obesity can be prevented through changes in lifestyle, nutrition, exercise, and other variables which can altering DNA transcription and consequent gene expression [9]. This is an epigenetic process and reflects the body’s interaction with the environment and its dynamic and adjustable molecular changes. This is why when we talk about obesity it is important to understand how lifestyle changes and different therapies can change gene expression and lead to favorable results regarding body mass index (BMI).

Recent years have seen the design of drugs to reduce BMI, and the nutraceutical market for weight loss has grown. Considering the many side effects that weight loss drugs have, nutraceuticals seem to be a much more suitable choice due to absence of toxicity. This review provides an overview of the intricate interplay between epigenetics and the most studied nutraceuticals, such as resveratrol, curcumin, epigallocatechin-3-gallate (EGCG), ginger, capsaicin, and caffeine, in the context of obesity management.

## 2. Methods

For this narrative review, we have summarized the articles that could be relevant, using the academic databases Pubmed and ScienceDirect for this purpose. Because we focused on the impact of nutraceuticals in epigenetics modulation in the obesity literature, a search was performed with the following search terms: obesity, monogenic obesity, polygenic obesity, epigenetic of obesity, nutraceuticals, resveratrol, curcumin, epigallocatechin gallate, ginger, capsaicin, and caffeine, which were used during the literature survey individually or in combination. We searched for clinical studies, original research, and reviews published in the English language until January of 2023. Due to the numerous published articles on epigenetics of obesity and nutraceuticals included in the study, as well as the limited number of references allowed, we focused on the most impactful and relevant papers supporting the hypothesis of obesity gene modulation by nutraceuticals.

## 3. Aspect of Genetic Obesity

From a genetic point of view, obesity is commonly classified into subgroups according to the presumed etiology: monogenic obesity, extremely severe but not accompanied by developmental delay; syndromic obesity, involving obese subjects with dysmorphia, mental retardation, and developmental anomalies; and polygenic obesity, also called common obesity [10]. However, the forms of obesity transmitted hereditarily are very rare and represent a small part of all cases of obesity present in the population. Syndromic obesity is caused by chromosomal rearrangements such as WAGR syndrome, Prader-Willi syndrome, Bardet-Biedl syndrome, SIM1 syndrome, Down syndrome, etc. [11].

The most frequent causes of monogenic obesity are mutations in the genes leptin (LEP), leptin receptor (LEPR), melanocortin 4 receptor (MC4R), proopiomelanocortin (POMC), proprotein convertase subtilisin/kexin type 1 (PCSK1), and neurotrophic receptor tyrosine kinase 2 (NTRK2) [12].

### 3.1. Common or Polygenic Obesity

Common obesity, defined by a BMI > 30 kg/m^2^, is a continuous quantitative trait. The genetics of quantitative traits in humans have historically made use of observational studies on pairs of twins. The first studies on the heritability of common obesity date back to the late 1970s, including one important study that observed more than 15,000 pairs of twin brothers, roughly 6000 of whom were monozygotic and 7500 of whom were dizygotic. This was a longitudinal study that followed a large group of war veterans over time (25 years), studying both concordance and heritability with the Falconer method for studying twins; the results showed that BMI, weight, and high were highly correlated across time.

In the last two decades, genome-wide association studies (GWAS) for obesity phenotype have shown a correlation between polymorphism in FTO and BMI, identifying two other polymorphisms associated with the phenotype that map in the proximity of the MC4R gene. It is interesting to note that in these cases a cumulative effect of two susceptibility variants was identified; subjects with risk-conferring alleles in both FTO and MC4R have a higher BMI compared to subjects with only one risk allele (FTO or MC4R) [13].

Several other studies have identified different susceptibility variants in different genomic regions and for different study populations, according to which dozens of genetic susceptibility variants are known, each of which provides a very modest contribution to the formation of the phenotype. Furthermore, studies of genes regulating glycolipid metabolism and thermogenesis revealed that a specific polymorphism (Pro12Ala) in the peroxisome proliferative activated receptor gamma (PPARγ) gene is associated with lower BMI and increased insulin sensitivity [14]. Similarly, recent meta-analyses involving up to 7000 subjects have demonstrated a significant association between BMI and two other polymorphisms, the Trp64Arg SNP in the β3-adrenergic receptor (ADRB3) gene [15] and the insertion/deletion (I/D) in the Uncoupling protein-2 (UCP-2) gene [16]. In contrast, the results regarding the impact of the 2G866A SNP in the UCP-2 gene on obesity remain inconclusive.

Therefore, polygenic obesity results from the complex interaction between genes and the surrounding environment, physical activity, diet, and gender [17].

### 3.2. The Role of Epigenetics in Obesity

As we have seen, the excessive increase in obesity in recent decades is impossible to explain only through genetics, which is why studies have turned to epigenetics. Certain variables, for example nutrition and physical exercises have the ability to modify gene expression without affecting the DNA sequence, a phenomenon known as epigenetics. The term “epigenetics” was introduced in 1942 by the biologist Conrad Hal Waddington, referring to certain inherited changes not accompanied by changes in the DNA sequence itself. The most important and studied epigenetic mechanisms are DNA methylation, histone modification, and non-coding RNAs (ncRNAs), which can be transmitted transgenerationally through mitotic or meiotic cell division [18]. Epigenetic programming of parental gametes, the fetus, and early postnatal development can be influenced by environmental factors; therefore, even gene expression can be altered in response to environmental exposures. The animal model of agouti mouse with the agouti viable yellow (*A^vy^*) mutation is a classic example of obesity that can be modulated through epigenetic mechanism [19]. The *Agouti* gene in mice controls the color of hair, and is under the control of a specific promoter in exon 2. During hair follicle cell development, the gene switches ON at a specific time and produces an agouti coat with a yellow stripe in the dark hair. The mutant *A^vy^* form of this gene makes the mice yellow and predisposes them to obesity, diabetes, and cancer. However, when the mutant pregnant mice are fed food enriched with extra methyl groups, non-obese brown pups resulted that were longer lived [3,20].

Following this example, later results showed that the mutant *Agouti* gene in the obese yellow mice is unmethylated and turned on, while in the brown mice the gene is methylated and shut down. The color of the agouti mice acts as a sensor for the DNA methylation status in the color gene. The agouti mice model allows us to understand how environmental factors and maternal diet affect the epigenome and regulate gene expression.

#### 3.2.1. DNA Methylation

However, one of the main mechanisms responsible for this process is DNA methylation capable of activating or deactivating a determined gene, thereby suppressing or activating the relative function [21]. DNA methylation changes can be dynamic in the sense that they can be modified under the influence of environmental factors or can be stable and transmitted to the next generations [22]. In mammals, 98% of DNA methylation occurs in a CpG dinucleotide context in somatic cells and above a quarter appears in a non-CpG context in embryonic stem cells. DNA methyltransferases (DNMTs), DNA demethylases, and the ten–eleven translocation (TET) proteins are involved in the regulation of DNA methylation. In mammals, five family members have been described: DNMT, DNMT1, DNMT2, DNMT3a, DNMT3b, and DNMT3L, of which only DNMT1, DNMT3a, and DNMT3b possess DNMT activity. DNMT1 maintains DNA methylation, while DNMT3a and DNMT3b can establish new DNA methylation. Removal of DNA methylation is mediated by TET proteins (TET1, TET2, and TET3) [23].

Certain human exome sequencing studies have identified de novo mutations in DNMT3A in subjects with autism spectrum disorder. Mice with whole-body DNMT3A haploinsufficiency show obesity in adults life associated with the autism phenotype [24]

DNA methylation can be strongly influenced by diet. Certain nutrients, such as folate, methionine, choline, vitamin B12, and pyridoxal phosphate can influence DNA methylation status, in this manner playing an important role in the modulation of gene expression, in particular of genes connected to main metabolic regulations such as energy balance and body composition [25]. Regarding the correlation between genes involved in polygenic obesity and methylation, several studies confirm the link between the methylation state of certain genes and obesity. A recent cohort study found a negative association between hypomethylation of the LEP gene promoter, obesity, lipid profile modification, and low insulin sensitivity [26]. In another cross-sectional study involving obese pre-bariatric adult subjects, LEP gene methylation was negatively correlated with BMI, while ADIPOQ adiponectin methylation resulted in a positive association [27]. The association between obesity and methylation of the LEP and ADIPOQ genes has been confirmed by several studies [28,29,30].

Methylation of PGC1A (peroxisome proliferator-activated receptor γ coactivator 1 alpha), a transcription factor critical in energy expenditure, and of IGF-2 (insulin-like growth factor 2), was disrupted in high fat diet (HFD), gestational diabetes, and obesity, with caloric restriction restoring the methylation state [31].

The NPY gene stimulates food intake, while the POMC gene promotes satiety. hypermethylation of POMC and hypomethylation of NPY was found in obese people [32]. Certain genes involved in inflammation and oxidative stress, such as TNF, TFAM (mitochondrial transcription factor A), and Il6, showed aberrant methylation levels in obese people [33]. In subjects with obesity and metabolic diseases, a hypermethylation of the IRS1 promoter and of the PIK3R1 gene was noted in obese individuals [34].

#### 3.2.2. Histone Modifications

Histones are well-conserved proteins involved in DNA packaging into chromatin. The most common and well-known histone modifications are acetylation, methylation, phosphorylation, O-GlcNAcylation, adenosine diphosphate (ADP) ribosylation, and lactylation. Modifications to the histone tails change the chromatin structure and regulate enhancer and promoter activities [35,36]. Acetylation opens the chromatin structure, favoring the binding of factors necessary for gene transcription; hyperacetylation is associated with an active transcription state, while histone deacetylation is associated with compacted and condensed chromatin, causing transcriptional repression. Histone acetylation and deacetylation are catalyzed by histone acetyltransferases (HATs) and HDACs, respectively. HATs catalyze the transfer of the acetyl group from acetyl-CoA onto a lysine residue, which can be reversed by HDACs [37].

Histone methylation can occur at the level of lysine and arginine residues [38]. Histone methylations may confer either active or repressive transcription depending on their positions and methylation states. H3K4, H3K36, and H3K79 methylations are markers of active transcription, whereas H3K9, H3K27, and H4K20 methylations are associated with suppressive transcription. These histone methylations are regulated by histone methyltransferases (HMTs, ‘writers’) and histone demethylases (HDMs, ‘erasers’). The methylation reaction can be reversed via multiple mechanisms [39]. PPARγ and C/EBPα are considered the most important regulators of adipogenesis, both in culture and in vivo, followed by C/EBPβ and C/EBPδ (which are essential for adipose tissue development), preadipocyte factor-1 (Pref-1), and adipocyte protein 2 (aP2). All these genes are regulated by histone modification during adipocyte differentiation [40]. In addition, histone modifications are involved in the control of the appetite-regulated genes POMC and NPY. Investigation of histone tail modifications on hypothalamic chromatin extracts from 16-day-old rats showed decreased acetylation of lysine 9 in histone 3 (H3K9) for the POMC gene and increased acetylation for the same residue for the NPY gene, modifications correlated with altered expression of the genes and obesity [19,41]. In the liver of obese mice fed an HFD, an increase of H3 lysine 9 and 18 acetylation was observed at TNF*α* and Ccl2 (monocyte chemotactic protein 1). On the other hand, weight loss interventions increased H4 acetylation at the GLU4 gene, increasing its expression [42]. All these findings show that histone modification can play an important role in obesity [43].

#### 3.2.3. Non-Coding RNAs

MicroRNAs (miRNA) are small non-coding RNA molecules with an average length of 22 nucleotides. Despite their small size, they are able to control gene expression by exploiting the perfect or similar complementarity they have with other RNA molecules [44]. Recent findings have shown that miRNAs regulate adipogenesis and are involved in obesity development. The miRNAs involved in adipogenesis are *miR-30*, *miR-26b*, *miR-199a*, and *miR-148a.* In obese subjects fed a high fat diet, researchers have found high levels of this miRNA. In obese adults, *miR-17-5p* and *miR-132* were more highly expressed in the visceral adipose tissues and were correlated with impaired glucose, high BMI, and dyslipidemia [45].

*miR-26b* is involved in the adipogenesis process, and studies regarding the proliferation of human preadipocytes that overexpress *miR-26b* exhibited increased triglyceride content in the adipocytes and upregulation of PPAR-γ expression during differentiation [46,47].

*miR-200a*, *miR-200b*, and *miR-429* are upregulated in the hypothalami of obese and leptin deficient ob/ob mice. Treatment with leptin downregulates these miRNAs, and hipotalamic silencing of *miR-200a* increases expression of LEPR and ISR-2 (insulin receptor substrate 2), suggesting that this miRNA can be a target for obesity treatment [48].

In the last decades, a large number of miRNAs differently expressed in obese people have been discovered involved in fat metabolism, adipogenesis, hypoxia, insulin signaling, inflammation, and cell differentiation and development. A comprehensive list of miRNAs related to obesity and metabolic diseases was published by Landrier et al. [45].

## 4. Nutraceuticals

The term “nutraceutical” was coined by Stephen L. De Felice, who used it for the first time in a 1989 publication; it comes from the union of the words “nutrition” and “pharmaceutical” [49]. Since then, it has been used to indicate the science that studies the individual components of foods that have health benefits [50]. Our ancestors already understood that eating certain types of food had consequences on the body. “Let food be your medicine and your medicine be food” is a phrase attributed to Hippocrates, the father of scientific medicine, who already 400 years before the birth of Christ understood the link between nutrition and well-being. Today the food nutrients that have the power to bring benefits to our body are studied biologically and chemically, and through nutraceuticals it is possible to know how they act in our body.

Regarding weight loss, nutraceuticals have many ways to act; as nutrient absorption regulators (ginseng, green tea, chitosan, psyllium, inulin, guar gum), appetite regulators (whey proteins and chlorogenic acid), energy expenditure modulators (curcumin and L-carnitine), and on the fat metabolism (resveratrol and flaxseed) [51].

Nutraceuticals are not involved in primary metabolism, and usually have health-promoting bioactivities. According to computational chemistry data, there are about 400,000 bioactive compounds of pharmaceutical interest [52]. Obesity is a very complex diseases involving the coexistence of different signals, including epigenetic changes [53]. For this reason, an ideal nutraceuticals formula should be able to act on all possible pathways regarding obesity, which we can expect to be very challenging.

In this review, we describe nutraceuticals that have been shown to be helpful in obesity by modulating gene expression. We focus on the most studied substances in this area, such as resveratrol, curcumin, ginger, EGCG, capsaicin, and caffeine, as shown in Figure 1. The most important epigenetic mechanisms through which these nutraceuticals act beneficially in obesity are summarized in Appendix A.

### 4.1. Resveratrol

Resveratrol (3,5,4′-trihydroxystilbene) is part of a large group of phytochemicals, including flavonoids and lignans, and is produced by plants in response to attack by pathogens such as bacteria or fungi as well as to injury [54]. Sources of resveratrol in food include grapes (*Vitis vinifera* L.), raspberries (*Rubus idaeus* L.), blueberries (*Vaccinium corymbosum* L.), peanuts (*Arachis hypogaea* L.), and mulberries (*Morus alba* Hort. ex Loudon L.) [55,56]. Resveratrol presents two geometric isomers, cis-(Z) and trans-(E). The *trans* form can undergo isomerization to the cis form when exposed to ultraviolet radiation [57]. In prevalence and biological activity, the dominant form is cis [58,59]. Resveratrol was first isolated in 1940 from the roots of white hellebore (*Veratrum album* L.) [60] and subsequently from knotweed in 1963 (*Polygonum cuspidatum* Sieb. et Zucc.), which is one of the richest sources of resveratrol in nature and plays an important role in ancient Chinese and Japanese medicine [61]. Resveratrol has been demonstrated in translational models to have anti-obesity and metabolic reprogramming properties [62,63]. It has been shown to be beneficial in animal studies for reducing body weight, insulin resistance, adipose tissue size, and weight [64,65].

The mechanisms through which resveratrol exerts its beneficial effects seem to be related to gene expression modulation and changes that mimics calorie restriction [66]. In patients with obesity, intracellular targets such as the deacetylating enzyme sirtuin-1 (SIRT-1), adenosine monophosphate-activated protein kinase (AMPK), and peroxisome proliferator-activated receptor γ coactivator-1α (PGC-1α) are altered [67]. In this context, resveratrol can activate the SIRT-1 gene, which plays an important role in mitochondrial activity modulation, glucose homeostasis, and other metabolic conditions related to obesity [68]. 

Numerous studies have analyzed the effect of resveratrol on adipogenesis using both in vivo and in vitro models, finding that resveratrol has an inhibitory effect on this process. Thus, the incubation of pre-adipocyte cells in vitro with different concentrations of resveratrol (1, 10, and 25 µM) for 24 h showed a lower expression of acetyl-CoA carboxylase (ACC), a decrease in the content of triacylglycerol, and inhibition of lipoprotein lipase [69]. Authors have found that resveratrol is able to reduce adipogenesis by downregulating the expression of CCAAT-enhancer-binding protein (C/EBPα), sterol regulatory element-binding protein 1c (SREBP-1c), and PPARγ after the incubation of pre-adipocytes with >10 μM resveratrol concentration [70]. Other authors confirmed these findings using (10–40 μM) resveratrol dosage and incubation times of 2, 4, and 6 days; the best result was achieved at 40 μM resveratrol dosage and 6 days incubation time, when 40% of pre-adipocyte differentiation was inhibited [71].

Several studies have demonstrated that resveratrol can increase expression of the thermogenesis markers UCP1 and BMP7; followed by reduction of fat accumulation, increased oxygen consumption and due to these processes exhibits an important effect on thermogenesis and the browning process [72]. Fibronectin type III domain-containing protein 5 (FNDC5) can promote conversion of white adipose tissue (WAT) to brown adipose tissue (BAT) by increased UCP1 expression [73], and it was demonstrated that resveratrol can increase FNDC5 expression in the subcutaneous adipose tissue from mice and humans [74].

In another study, oral administration of resveratrol in mice fed with a standard diet improved glycemic and lipidic profile and enhanced thermogenesis by increasing UCP1 expression in BAT. The authors suggested that increased expression of UCP1 was associated with increased expression of SIRT-1, PTEN, and BMP7 in the same BAT [72].

In addition, resveratrol fights against obesity by enhancing catecholamine production, suppressing pro-inflammatory M1 macrophages and activation of anti-inflammatory M2 macrophages in WAT [75]. In mice fed HFD resveratrol by activation of PI3K/SIRT1 and Nrf2 signaling pathways, the inhibition of transcriptional regulators (e.g., EP300 gene) decreased fat mass and body weight, modulated insulin and glucose metabolism, and restored immune dysfunction [76,77]. Being able to protect against sarcopenic obesity through the PKA/liver kinase B1 (LKB1)/AMPK pathway, resveratrol improves mitochondrial function and reduces oxidative stress [78].

Administration of resveratrol in rat model with diet-induced obesity has been shown to reduce body weight, subcutaneous adipose tissue (SAT) masses, and WAT. Regarding the effects in microRNA expression after resveratrol administration, *miR-211-3p*, *miR-1224*, and *miR-539-5p* were increased and *miR-511-3p* was decreased. The target genes were PPAR-γ, SP1 transcription factor (SP1), and hormone-sensitive lipase (HSL), which are involved in FA (fatty acids) metabolism in adipose tissue [79]. The SP1 gene is a target of *miR-1224* and *miR-539-5p*, both of which are involved in SP1 regulation. The synergistic action of SREBP-1 and SP1 induces the expression of FAS, which promotes de novo lipogenesis [80]. The expression of SP1 and SREBP-1 was downregulated by resveratrol due to upregulation of *miR-539-5p*.

Resveratrol can act as a prebiotic as well, being metabolized by gut microbiota, and produces metabolites such as lunularin and dihydroresveratrol [81]. In mice fed HFD, resveratrol intake of 200 mg/kg/day had anti-obesity effects by improving gut dysbiosis, *Bacteroidetes/Firmicutes* ratio, increasing the abundance of *Lactobacillus* and *Bifidobacterium*, and inhibiting the growth of *Enterococcus faecalis* [65,82].

A positive correlation was found between body weight and *Enterococcus faecalis* and a negative one with *Lactobacillus*, *Bifidobacterium*, and *Bacteroidetes*/*Firmicutes* ratio. Resveratrol intake in these mice increased FIAF and decreased LPL gene expression, suppressing fatty acid biosynthesis in the liver [82].

The recommended doses of resveratrol are 250–1000 mg orally daily for up to three months. Overall, it is well tolerated in healthy individuals; however, adverse effects, including nephrotoxicity and gastrointestinal symptoms such as nausea, diarrhea, and abdominal discomfort, have been reported in human subjects. In doses higher than 1000 mg/day, it has been reported that resveratrol modifies the activity of cytochrome P450 isoenzymes, leading to interactions with other medications. [83,84].

Due to all the effects described above, resveratrol can be used as a therapeutic agent against obesity. Of course, due to data scarcity, further studies remain needed to evaluate the long-term effects of resveratrol, especially in vivo and in human trials.

### 4.2. Curcumin

*Curcuma longa* L. is a perennial and rhizomatous herbaceous plant which belongs to the family of Zingiberaceae, as ginger (*Zingiber officinale* Rosc.) does. The most important component of nutritional and phytotherapeutic interest is the root, constituted by a cylindrical, branched, aromatic rhizome of orange-yellow color. In traditional Indian, Thai, and Middle Eastern cuisine it is used in food as a spice. The term curcumin usually refers to 1,7-bis(4-hydroxy-3-methoxyphenyl)-1,6-heptadiene-3,5-dione, a compound known as “curcumin I”, although the plant contains more than 100 chemical compounds. The two other best-known compounds are curcumin II (demethoxycurcumin, 1-(4-hydroxy-3-methoxyphenyl)-7-(4-hydroxyphenyl)-1,6-heptadiene-3,5-dione) and curcumin III (bisdemethoxycurcumin, 1,7-bis(4-hydroxyphenyl)-1,6-heptadiene-3,5-dione) [85]. The yellow color is due to “curcumin I” and the curcuminoids bisdemethoxycurcumin and demethoxycurcumin, generally used as a natural dye and in the food industry [86]. Turmerone (ar-turmerone), β-turmerone, α-turmerone, β-bisabolene, β-sesquiphellandrene, α-zingiberene, curcumol, and curcumenol are the principal essential oils of curcumin [87]. Curcumin has several biological activities, and functions as an antioxidative, anti-inflammatory, anti-cancer, and anti-obesity agent.

Regarding obesity, it has been shown that curcumin can interfere with adipocyte differentiation [88] and alter the adipocyte life cycle. Curcumin has anti-adipogenic functions and can suppress the 3T3-L1 adipogenesis in murine cell models and in human primary preadipocytes by stimulating the Wnt signaling cascade [89,90,91]. During adipocyte differentiation, curcumin blocks mitotic clonal expansion by inhibition of transcription factors such as C/EBPα, Krüppel-like factor 5(KLF5) and PPARγ [91,92]. Adipocyte protein 2 (aP2) is a mature adipocyte marker, and in the 3T3-L1 cells it was found that curcumin decreases aP2 microRNA expression, increases expression of certain Wnt pathways targets such as c-Myc and cyclin D1, and reduces mitogen-activated protein kinase (MAPK) phosphorylation, which is associated with differentiation of 3T3-L1 cells into adipocytes [92].

In several clinical trials, oral curcumin intake decreased the plasma lipid levels induced by HFD, an aspect correlated with AMPK and PPARα activation (both of which inhibit acetyl-CoA carboxylase (ACC)), followed by a decrease in lipid accumulation and FA synthesis [93,94]. In animals with hyperlipidemia induced by a high glucose diet (HGD), curcumin intake resulted in reducing total cholesterol, triglycerides, fatty acids [95,96]. and transaminase levels, along with insulin resistance improvements [97,98]. Therefore, in obese subjects, curcumin can reduce inflammation and insulin resistance by inhibiting activation of signal transducers and activation of transcription 3 (STAT3) in human adipocytes [99].

In the C57BL/6 mice on a regular low-fat chow diet receiving curcumin gavage (50–100 mg/kg body weight per day) for 50 days, a browning of WAT was observed, which is associated with an increase of gene expression involved in thermogenesis and mitochondrial biogenesis. Curcumin can increase body temperature, energy expenditure, and UCP1 expression in the BAT [100].

After oral administration, curcumin is distributed throughout the intestines, where it affects the GM composition [101]. Studies on animal models have shown that curcumin intake improve the richness of intestinal microbiota and has significant effects on family members such as *Bacteroidaceae*, *Prevotellaceae*, and *Rikenellaceae* [102]. In addition, curcumin improves the composition of gut microbiota in colitis mice [103]. In a menopausal rat model, curcumin administered 100 mg/kg per day had a weight-loss effect by improving gut dysbiosis and promoting species such as *Anaerotruncus*, *Exiguobacterium*, *Helicobacter*, *Shewanella*, and *Serratia* [104]. Islam et al. [105] found that administration of curcumin for 14 weeks in a human-equivalent dose of 2 g daily as supplementation in a HFD was correlated with reduced adiposity and relative abundance of *Lactococcus*, *Turicibacter*, and *Parasutterella* genera. Scientists believe that curcumin has protective effects in dietary obesity due to downregulation of inflammation in adipose tissue. Curcumin seems to attenuate the high-fat and high-cholesterol Western-type diet (WD)-induced chronic inflammation and associated metabolic diseases, including obesity. Furthermore, by reducing the dysfunction of the intestinal barrier, curcumin modulates chronic inflammatory diseases despite its low bioavailability. The Western diet is characterized by high fat and refined carbohydrates intake, which induce metabolic diseases and inflammation. Curcumin pretreatment in this case is associated with reduced IL-1β-induced activation of p38 MAPK in intestinal epithelial cells (IECs) [106].

To discover new therapeutic targets against obesity, it is necessary to understand the mechanism of adipogenic differentiation. Adipogenic differentiation is repressed by activation of Wtn pathway, and certain microRNAs intervene in the regulation of pre-adipocyte differentiation and proliferation. An important component of the Wtn signaling cascade is TCF7L2, which together with free β-catenin (β-cat) molecules forms the complex transcription factor β-cat/TCF, a key effector of Wtn [107]. TCF7L2 is one of the most studied genes involved in the development of type 2 diabetes [90,108]. Silencing of TCF7L2 leads to impaired adipogenesis, disruption of Wnt signaling, and upregulation of axis inhibition protein 2 (Axin2) mRNA [109]. Tian et al. found that curcumin treatment attenuated *miR-17-5p* expression and stimulated TCF7L2 expression in 3T3-L1 cells [107].

Turmeric is safe when used short-term, having been approved by the US Food and Drug Administration (FDA) as “Generally Recognized as Safe”. For example, a maximum of 8 g of curcumin per day is safe when used for up to 2 months. Administration of up to 3 g per day appears to be safe when used for up to 3 months. Turmeric can be used in doses of up to 1.5 g daily for up to 9 months. Generally, turmeric does not cause serious side effects; however, when taken in large doses or for extended periods it may cause gastrointestinal symptoms such as diarrhea, nausea, and stomach upset [83,110].

Unfortunately, for now it is unknown how curcumin treatment represses adipogenesis in vivo and reduces body weight gain. As can be seen, curcumin is a promising compound in obesity prevention and reduction, and there are many preparations containing this nutraceutical. However, more comprehensive human studies are necessary in order to understand how exactly to use it in obesity management.

### 4.3. Ginger

*Zingiber officinale* Roscoe is a perennial herbaceous plant native to Asia and India. In the first century it was introduced to the Mediterranean area, and in the third to Japan. In England and America, it arrived in the eleventh century. Today, it is mostly cultivated in Africa and Asia. Ginger is a spice and flavoring agent for food, being used in cuisine in different forms such as fresh, dry, oil, paste, and emulsion [111]. Ginger is rich in bioactive phenolic compounds such as gingerols, shogaols, paradol, and zingerones. Ginger has been used since antiquity in Ayurvedic, Chinese, and Yunani medicine to treat nausea, rheumatoid arthritis, muscular aches, indigestion, sore throats, constipation, and fever, and is a stimulant and carminative of the gastrointestinal tract [112,113]. The health potential of ginger has been intensively studied, and different regulatory authorities consider it a safe nutraceutical [114].

Recently, the anti-obesity effects of ginger and its compounds have been taken into consideration, and the results from in vitro and in vivo studies support this idea [115,116,117,118,119,120], although research in humans remains limited. Ginger seems to be able to influence body composition and weight through several mechanisms, such as inhibition of adipocyte differentiation and lipid accumulation [117] and through increasing thermogenesis, lipolysis, and energy expenditure [115,120].

Ginger can interfere with adipogenesis of 3T3-L1 cells. It is known that the 3T3-L1 cell line is the most reliable model for research on adipogenesis [121]. In several studies it has been shown that 6-gingerol ((S)-5-hydroxy-1-(4-hydroxy-3-methoxyphenyl)-3-decanone), one of the most abundant components in ginger root, was able to inhibit adipogenesis via different mechanisms. In one study, 6-gingerol diminished the insulin-stimulated serine phosphorylation of Akt (Ser473) and GSK3 (Ser9) and suppressed adipogenesis by down-regulation of C/EBP and PPARγ followed by subsequent inhibition of the expression of relevant markers for lipid accumulation, aP2, and FAS. In addition, 6-gingerol can suppress differentiation of 3T3-L1 cells by attenuating the Akt/GSK3 pathway [122].

Li et al. found that 6-gingerol can inhibit adipogenesis by down-regulation of C/EBP-α and PPAR-γ followed by inhibition of FAS and Acetyl-CoA carboxylase (ACC) expression and activation of the canonical pathway Wnt/β-catenin, the activation of which is responsible for dephosphorylation and nuclear translocation of β-catenin. 6-gingerol increases the mRNA and protein levels of dishevelled segment polarity protein 2 (DVL2) and Low-Density Lipoprotein Receptor-Related Protein 6 (LRP6), an important component of Wnt, suggesting that up-regulation of several components of the Wnt/β-catenin pathway results in inhibition of adipogenic differentiation [123].

6-Shogaol [1-(4-hydroxy-methoxyphenyl)-4-decen-one], another important constituent in ginger, has the ability to inhibit adipogenesis of 3T3-L1 preadipocytes at 40 µM and to decrease expression of adipogenic/lipogenic markers levels such as PPAR-ƴ, FAS and C/EBPꭤ. By increasing glycerol release, 6-shogaol decreases intracellular lipid accumulation [124]. This compound decreases the phosphorylation of IRS-1, PI3K, and AKT, suggesting that 6-shogaol exerts its antiobesity effect through the PI3K/AKT pathway [125].

The antiobesity effects of 6-gingerol and 6-shogaol seems to be associated with their anti-inflammatory properties. These compounds can downregulate mRNA levels of tumor necrosis factor alpha (TNFα) and interleukin-6 (IL-6) in the adipose tissue of rats fed HFD [126,127]. Studies have shown that both compounds are able to inhibit TNF-mediated downregulation of adiponectin expression in adipocytes by functioning as PPARƴ agonists. 6-gingerol inhibits the phosphorylation of anti-phospho-c-Jun-NH2-terminal kinase (JNK) and activates the upstream kinase of JNK. These aspects are very important and suggest that the ginger compounds might have important implications in diabetes prevention via improvement of adipocyte dysfunction [128]. To date, several studies have demonstrated the beneficial effects of ginger compounds in regulation of glycolytic enzyme, limitation of gluconeogenesis, and insulin sensitivity improvement in diabetic animal models [129,130].

6-gingerol protects pancreatic cells from oxidative stress and shows significant anti-inflammatory properties in amelioration of insulin resistance in fructose-induced adipose tissue, probably by suppressing adipose macrophage. By controlling inflammation and inhibition of TNFα and IL6 expression, 6-gingerol exerts a hypoglycemic effect [131,132,133].

Another ginger compound, 12-dehydrogingerdione, has anti-inflammatory and antioxidant properties, and stimulated nitric oxide (NO) production and suppressed mRNA levels of cytokines IL-6, IL-8 in RAW 264.7. In addition, ginger extract can reduce the intestinal absorption of carbohydrates with antihyperglycemic effect [134,135,136].

In a recent study, it has been shown that ginger supplementation (5% ginger flour intake for 7 weeks) can ameliorate metabolic parameters induced by HFD. C57BL/6 mice fed a HFD ginger supplementation showed body weight loss and improvements in lipid and glucose profiles. By upregulation of the genes involved in fatty acid oxidation, such as fibroblast growth factor 21 (FGF21), carnitine palmitoyltransferase 1 (CPT1), and acyl-CoA oxidase 1 (ACOX1), ginger supplementation determined a reduction in the accumulation of hepatic lipids. In addition, an upregulation of antioxidant enzymes involved in the first line of defense against oxidative damage, such as nuclear factor erythroid 2-related factor (NRF), 1/2 superoxide dismutase (SOD), and glutathione peroxidase (GPX), was observed [137].

Adipocyte size was reduced by ginger supplementation, which was accompanied by PPAR and aP2 attenuation. In contrast, CPT1 gene expression was upregulated by ginger administration [138].

It has been demonstrated that ginger can reduce body weight, with positive effects on HDL cholesterol level and increase of peroxisomal catalase level [139]. In addition, ginger can decrease appetite by its modulatory effect on 5-hydroxytryptamine [120].

Regarding the effect on thermogenesis and energy expenditure, ginger seems to have an agonistic effect on transient receptor potential vanilloid Type 1 (TRPV1). This is a temperature sensor able to activate the sympathetic nerve system and norepinephrine (NE) release, with a resulting increase of uncoupling protein-1 (UCP-1) expression, thermogenesis followed by cAMP high production, increased activity of hormone sensitive lipase (HSL), and lipolysis [140].

Uncoupling protein 1 genes play an important role in body weight regulation and energy homeostasis, being associated with resistance to weight loss, reduced resting energy expenditure, and increased weight gain over time [141]. Ginger supplementation significantly decreased body weight, waist circumferences, hip circumference (HC), and waist-to-height ratio (WHtR) in the G allele carriers of UCP1 (-3826A>G polymorphism and Arg alleles carriers for ADRB3 (Trp64Arg) polymorphism [142].

In rats fed HFD and supplemented with high-hydrostatic pressure ginger extract (HPG) and hot water extract of ginger (WEG), a reduction of *miR-21* expression was observed followed by amelioration of adipogenic gene expression. *miR-132* levels were decreased in HPG supplemented rats, followed by a decrease in expression of inflammation-related genes such as Il6, MCP-1, and TNFα [143].

Ginger supplementation can modulate the composition of gut microbiota and the abundance of microbial taxa [144,145]. Short-term ginger juice supplementation increased the *Firmicutes*/*Bacteroidetes* ratio and the abundance of *Proteobacteria* in healthy adults and decreased pro-inflammatory *Ruminococcus* [146]. All these data suggest that ginger is able to decrease body weight and adipose tissue by modulating gene expression involved in these processes and could be useful as a nutraceutical for the prevention of obesity and inflammation.

Ginger can be taken orally in doses of 0.5–3 g per day for up to 12 weeks. While generally considered safe for most people when consumed in moderate amounts, excessive or prolonged use may have potential adverse effects. Doses higher than 5 g per day increase the risk of mild side effects, such as heartburn, diarrhea, belching, and general stomach discomfort. Cases of arrhythmias and low blood pressure have been cited as well. By increasing bile acid secretion, ginger can aggravate gallstone formation [83,147].

### 4.4. Epigallocatechin-3-Gallate (EGCG)

EGCG is the most abundant catechin in green tea (*Camellia sinensis* L.) and is found in smaller quantities in other foods such as carob (*Ceratonia siliqua* L.) flour, blackberries (*Rubus plicatus* L. Weihe & Nees), apples (*Malus domestica* (Suckow) Borkh.), raspberries (*Rubus idaeus* L.), prunes (*Prunus domestica* L.), pistachios (*Pistacia vera* L.), peaches (*Prunus persica* (L.) Batsch), and avocados (*Persea americana* Mill.) [50].

During the manufacturing process, tea leaves are heated to inactivate the enzymes, rolled, and dried. The process prevents oxidation of the constituents and stabilizes tea compounds during storage. Green tea is rich in polyphenolic compounds called catechins, such as epicatechin, epigallocatechin, EGCG, epicatechin gallate (ECG), and other polyphenols in low quantities, such as kaempferol, myricetin, quercetin, certain alkaloids, theobromine, and caffeine. In brewed green tea, catechins account for 30–42% of the dry weight. A green tea beverage (250 mL hot water and 2.5 g leaves) contains about 260 mg of catechins, of which about 96 mg is EGCG [148,149].

EGCG is a polyphenol with anti-inflammatory, antioxidant, and anti-obesity actions. As anti-obesity agent, it decreases weight gain and adipose tissue weight by decreasing calorie intake and AMPK activation in the liver, white adipose tissue, and skeletal muscle, reducing the absorption of lipids, cholesterol, triacylglycerols (TAGs), and leptins, stimulating energy expenditure and fat oxidation, and increasing the level of high-density lipoproteins and fecal excretion of lipids [150,151,152].

In HFD mice, 100 mg/kg daily EGCG supplementation decreased the expression of the genes ACC1, SCD1, FAS, PPARγ, C/EBPΒ, and SREBP1, all of which are involved in de novo synthesis of fatty acids, and increased the expression of the genes HSL, ATGL, PPARa, ACO2, and MCAD, which are associated with lipolysis and lipid oxidation in epididymal adipose tissues. An increase in the expression level of the genes PGC1ꭤ and aP2, which are associated with thermogenesis and fatty acid transfer, was observed as well [152]. Likewise, in mice fed HFD, treatment of EGCG increased the expression of genes involved in fat oxidation, such as MCAD, PPARGꭤ, UCP3, and NRF1 [153]. It is well known that UCP3 gene mutations are associated with increased risk of morbid obesity, dyslipidemia, and diabetes [154]. In the skeletal muscle of subjects characterized by obesity and insulin resistance, the gene expression of NRF1 and UCP3 are decreased. MCAD gene deficiency is associated with the most common fatty acid oxidation disorder, and enhancing its expression can increase fat oxidation [155]. By increasing expression of these genes, EGCG enhances basal metabolism and lipid oxidation and decreases body weight in HFD-fed mice. Similar results were obtained in obese Beagle dogs, in which supplementation with green tea extract increased PPARꭤ, GLUT4, and LPL expression, followed by insulin sensitivity and lipid profile improvement, suggesting that EGCG is able to reverse obesity-related metabolic disturbances [156].

Numerous epidemiological studies have demonstrated that circadian misalignment in human can be associated with metabolic syndrome, including obesity, hypertension, and insulin resistance [157]. In a mouse model, mutations in the genes CLOCK, Per2, Bmal1, and RORα were associated with various metabolic disorders, suggesting the importance of clock genes in metabolic regulation [158,159]. In C57BL/6J mice fed a high-fat high-fructose diet (HFHFD), supplementation with EGCG showed beneficial effects on circadian misalignment, obesity, insulin resistance, and lipid metabolism by normalizing the expression of the clock genes CLOCK and Bmal1 and by regulating the levels of SIRT1 and PGC1α.

Moreover, EGCG decreased fatty acid synthesis in the liver, increased BAT energy expenditure, and prevented adipocyte hypertrophy [160]. In an HFD-fed dog model, ECCG administration in both low doses (0.25 g/kg BW) and high doses (0.50 g/kg BW) decreased BW gain and suppressed liver inflammation by decreasing the expression of the COX-2 and iNOS genes involved in the inflammatory process [161].

In the skeletal muscle of diabetic rats, oral gavage of 100 mg EGCG for 3 month decreased expression of dynamin-related protein 1 (DRP1 and Beclin1 due to down-regulation of the ROS/ERK/JNK-p53 pathway and amelioration of excessive muscle autophagy [162].

In a randomized double-blind crossover study with obese women without comorbidities, administration of 738 mg green tea resulted in decreased expression of *miR-1297*, *miR-373-3p*, *miR-192-5p*, *miR-1266-5p*, and *miR-595* compared with a control group. These microRNAs are involved in the regulation of genes associated with the Coactivator Associated Arginine Methyltransferase 1 (CARM1), Transforming growth factor beta (TGF-beta), Bone morphogenetic proteins (BMPS), and ribosomal S6 kinase (RSK) pathways. This study showed the beneficial effects of green tea intake by suppressing the expression of microRNAs that target genes mainly involved in adipogenesis and carcinogenesis [163].

Liu et al. found that EGCG regulated the gut microbiome profile in HFD-fed mice by reversing the abundance of Bacteroides and Parasutterella species decreased by HFD, and also reduced *Allobaculum*, *Roseburia*, norank_*Erysipelotrichaceae*, norank_*Lachnospiraceae*, unclassified_f_*Ruminococcaceae*, *Anaerotruncus*, *Odoribacter*, *Enterorhadu*, and *Lachnospiraceae*_UCG_006 and induced the enrichment of *Akkermansia* [164]. In another study, EGCG treatment increased the abundance of the beneficial bacteria *Bacteroides*, *Christensenellaceae* and *Bifidobacterium* and inhibited the pathogenic *Fusobacterium varium*, *Enterobacteriaceae*, and *Bilophila* [165]. Beneficial effects on body weight were observed when adults consumed up to 460 mg/day, while doses higher than 800 mg/kg were responsible for adverse effects [83]. While EGCG is generally considered safe when consumed in moderate amounts through dietary sources, there have been concerns regarding its toxicity at high doses or in concentrated forms. Human and animal experiments suggest the liver is the primary target for toxicity [166,167].

All these data suggest that EGCG as a nutraceutical could be a therapeutic agent in obesity treatment and prevention via moderate intake. However, further investigations are needed, especially in humans, to confirm these hypotheses.

### 4.5. Capsaicin

Capsaicin is a chemical compound present in different concentrations in plants of the genus *Capsicum*, knows as chili pepper, the main source of capsaicinoids in nature. These chemical compounds are produced by plants as secondary metabolites as a way of defense to deter predators [168] and are responsible for the fruit’s burning sensation and spicy flavor [169]. The natural capsaicinoids are capsaicin (C), dihydrocapsaicin (DHC), homodihydrocapsaicin (HDHC), nordihydrocapsaicin (NDHC), and homocapsaicin (HC). The concentration of capsaicinoids in a pepper varies from species to species and from variety to variety, and act and stimulate differently [170]. Natural capsaicinoids are capable of exerting multiple pharmacological and physiological effects, and in clinical practice can be used for pain relief, cancer prevention, and weight loss [171].

The most abundant capsaicinoid in peppers is capsaicin (trans-8-methyl-N-vanillyl-6-nonenamide), discovered in 1846 by Tresh; its chemical structure was determined in 1919 by Nelson [172,173]. Over the years, several studies have demonstrated that from a pharmacological point of view capsaicin is one of the most important constituents currently used for treatment of pain syndromes and diabetic neuropathy [173].

In particular, capsaicin acts by binding to receptors present on the peripheral nerve endings, the so-called vanilloid receptors type 1 or TRPV1, mainly localized on polymodal sensory nerve fibers of type C. These play a fundamental role in hyperalgesia and allodynia. The analgesic effect of capsaicin following binding to these receptors is thought to involve the entry of calcium into the cell until the channel is closed; although until recently it was not clear how the loss of sensitivity was determined by the entry of calcium ions, we now know that it is related to a decrease in phosphatidylinositol 4,5-bisphosphate PIP2, which contributes to the desensitization of TRPV1 receptors [174]. TRPV1 is excited by capsaicin, and through the afferent nerves it contains, the resulting signal is transmitted to the spinal cord. The excitation of efferent nerves by the central nervous system causes an increased release of catecholamines (norepinephrine, epinephrine and dopamine) at the adrenal level. In turn, catecholamines bind β-adrenergic receptors, increasing thermogenic activity. At the gastrointestinal level, TRPV1 activation by capsaicin increases thermogenesis and activates UCP1 in BAT. Taking into account the exposed mechanisms, it is clear that capsaicin increases thermogenesis and energy expenditure via TRPV1 activation, as shown in Figure 2.

Capsaicin has shown anti-obesity effects, being able to induce body weight reduction, improve lipolysis in adipocytes, increase energy expenditure, increase satiety, and decrease the desire to eat [175]. The anti-obesity action of capsaicin might be linked to the activation of both UCP proteins and of the sympathetic nervous system. Both properties carry out an inducing activity towards the metabolism, increasing thermogenesis (Figure 2). For example, treatment for 12 weeks with 6 mg/day of oral capsinoids has been associated with the loss of abdominal fat and is conditioned by the presence or absence of certain genetic polymorphisms, in this case TRPV1 Val585Ile and UCP2 -866 G/A, which may be predictive of the type of therapeutic response [176]. TRPV1 has an important role in body metabolic health, including lipid and glucose metabolism, and its activation by capsaicin stimulates insulin secretion, increases GLP-1 level, and regulates glucose homeostasis [177,178,179]. In obese mice, capsaicin inactivated nuclear factor-κB (NF-κB) and activated PPARγ in a receptor-independent manner, with suppression of inflammatory response modulating the adipocyte function of adipose tissues macrophages, which are independent on TRPV1 [180].

Several studies have demonstrated that capsaicin inhibits the expression of leptin, PPARγ, and C/EBP-α while up-regulating adiponectin at the protein level, and in this manner efficiently induces apoptosis and adipogenesis to inhibit 3T3-L1 preadipocytes in vitro [181,182].

In C57BL/6 male obese mice fed HFD for 10 weeks, capsaicin supplementation improved glucose intolerance, decreased leptin and insulin concentrations, decreased TRPV-1 expression in adipose tissue, and increased adiponectin expression, accompanied with increased expression of PPARα and PGC-1α in the liver [183]. In C57BL/6J mice, capsinoid supplementation decreased BW gain and fat accumulation, increased energy expenditure by lipolysis activation, and increased cyclic adenosine monophosphate (cAMP) levels and PKA activity in BAT [184]. In addition, capsaicin can improve cholesterol level and counter the harmful effects of HFD in mice by increasing the expression of the thermogenic genes UCP-1, SIRT-1, BMP8b, and PGC-1α and by enhancing the respiratory exchange ratio [185].

Another study found that in HFD rats capsaicin treatment increased expression of oxidation and thermogenic genes in WAT. It was observed that expression of aldo-keto reductase (AKR1B7)-encoding mRNA was decreased in adipose tissues of obese mice and capsaicin treatment reversed expression. In humans, AKR1B1 is involved in development of diabetic complications. In the same study, capsaicin was able to decrease FABP4 and TNFα expression, and UCP2 expression level decreased upon HFD was normalized by capsaicin supplementation [186]

Baboota et al. [187] demonstrated that 3T3-L1 adipocytes treated with 1 µM of capsaicin increased the expression of PGC 1ꭤ, UCP1, BDNF, PRMD16, PPARꭤ, FOXC2, NCOA1, DIO2, and SIRT1, which are associated with browning of white adipocytes through a TRPV1-dependent mechanism.

Treatment with nonivamide, a capsaicin analogue, increased expression of the anti-adipogenic microRNAs mmu-let-7a-5p, mmu-let-7d-5p, and mmu-let7b-3p, which are associated with decreased PPARγ levels and anti-obesity effects [188].

In a recent study in HFD-fed mice, capsaicin supplementation was associated with weight loss and altered gut microbiota composition. Capsaicin increased the numbers of *Akkermansia*, *Bacteroides*, *Prevotella*, *Allobaculum*, *Odoribacter*, and *Coprococcus* and decreased the numbers of *Escherichia*, *Desulfovibrio*, *Sutterella*, and *Helicobacter*. In addition, capsaicin increased the abundance of SCFAs along with acetate and propionate concentrations, with positive effects in treatment and prevention of obesity [189].

Capsaicin is safe for short-term use. With long-term use side effects may occur, including stomach irritation, sweating, and runny nose [83]. It has been shown in various studies in which animals have been given high doses to have carcinogenic, neurotoxic, and genotoxic effects [190]. In humans, high consumption of chilis has been reported to be a risk factor for cancer of the upper gastrointestinal tract, possibly due to the irritating effect of capsaicinoids [191].

To summarize, capsaicin plays an important role in human health at the metabolic level, especially in obese people. It is a spice with a long culinary history, and as such its use to treat obesity is more feasible compared to other medical interventions. Nevertheless, a better understanding of its mechanisms of action is necessary, and additional research is required to determine the optimal dose and duration of capsaicinoids for weight loss.

### 4.6. Caffeine

1,3,7-trimethylxanthine, or caffeine, is an alkaloid found naturally in more than 60 plant species, including coffee beans (*Coffea arabica* L.), cocoa beans (*Theobroma cacao* L.), tea leaves (*Camellia sinensis* (L.) Kuntze), kola nuts (*Cola acuminata* (P.Beauv.) Schott & Endl., *Cola nitida* (Vent.) Schott & Endl.), guarana berries (*Paullinia cupana* Kunth), yerba mate (*Ilex paraguariensis* A.St.-Hil.), and guayusa (*Ilex guayusa* Loes.) [192,193]. 

Numerous epidemiological studies have linked caffeine consumption to health benefits in moderate coffee drinkers, including reduction of mortality [194,195], protection against cancer development [196], decreased risk of type 2 diabetes [197], controlling Parkinson’s disease [198], slowing the progression of dementia [199], and slowing the progression of many forms of liver disease [200]. In the sporting world caffeine is ubiquitous due to its ergogenic aids [201], and use of caffeine supplements has been shown to improve performance in such different aspects of exercise as muscular strength, muscular endurance, jumping, sprinting, and movement speed. In addition, caffeine ingestion before exercise in the morning seems to promote weight loss according to published studies [202]. Regarding the effect of caffeine on weight loss, several studies have demonstrated that caffeine intake promotes a decreasing in weight, BMI, and body fat [203].

Caffeine supplementation results in reduced food intake and increased energy expenditure by inhibition of phosphodiesterase-induced degradation of intracellular cAMP, increasing cAMP-dependent protein kinase (PKA), and inhibition of PI3K/AKT activity [204]. In another study, increased energy expenditure and reduced number of adipocytes were related to FAS activity inhibition and decreased expression of PPARƴ [205]. Caffeine was able to reduce lipid accumulation by increasing lipolysis in adipocytes and inhibiting insulin-stimulated glucose uptake [206].

Another study showed that caffeine supplementation inhibited the expression of C/EBPβ, C/EBPα, and PPARγ in 3T3-L1 preadipocytes during differentiation by regulating the expression of G1-S cell cycle markers [207].

In HFSD-fed rats, caffeine treatments (0.1%) equivalent to four cups of coffee in humans modulated the mRNA expression of pyruvate kinase (PKM), microsomal triglyceride transfer protein (MTTP), and FASN in the liver, resulting in reduced de novo fatty synthesis [205]. Velickovic et al. found that caffeine treatment increased UCP1 expression in mMSC culture and upregulated gene expression of PPARγ, FABP4, adiponectin, the beige markers CD137, CITED1, and P2RX5, and the brown-selective genes PRDM16, LHX8, PGC-1α, AR-ß3, and COX8b, following by conversion of white/beige cells into brown adipocytes and an increase in lipolysis [208].

Mitani et al. showed that incubation of 3T3-L1 adipocytes with caffeine for 24 h miniaturized lipid droplets and decreased accumulation by inhibiting expression of PPARγ and CCAAT/enhancer binding protein (C/EBP) α. Interestingly, the expression levels of C/EBPβ and C/EBPδ were not observed at the level of mRNA, only at the protein level. The same authors found that caffeine inhibited expression of SERPIN1 in Caco-2 cells and decreased secretion of the proinflammatory cytokines Il-8 and PAI-1 [209].

It has been demonstrated that caffeine ingestion improves hyperglycemia in KK-Ay type 2 diabetic mice, enhances insulin sensitivity, significantly decreases the expression of the inflammatory cytokines TNFα, MCP-1, and IL-6, and decreases the mRNA levels of the macrophage marker F4/80 in adipose tissue, suggesting that caffeine is able to lower the production of the inflammatory adipocytokines and decrease inflammation. In addition, caffeine intake improved dyslipidemia, fatty liver and total cholesterol serum level with decreasing in SREBP-1 and FAS genes expression. An increase in IRS- 2 expression was observed as well, which is correlated with an improvement in insulin resistance at the hepatic level [210].

Regarding the effects of caffeine on gut microbiota, studies have shown an improvement in *Firmicutes*/*Bacteroidetes* ratio and increase in *Bifidobacterium* spp. in humans [211]., High fecal levels of *Bacteroides*-*Prevotella*-*Porphyromonas*, which are correlated with better metabolic status, have been observed in high coffee consumers [212,213].

Caffeine is administered orally in doses of 50–260 mg daily. It is safe for most healthy adults when used in doses up to 400 mg per day, the equivalent of about 4 cups of coffee. When used over a long period of time or in doses over 400 mg per day, caffeine can cause insomnia, nervousness, restlessness, nausea, increased heart rate, and other side effects. Higher doses can cause headaches, anxiety, and chest pain [83]. Caffeine overdose may cause hypokalemia, hyponatremia, impaired iron and zinc absorption, rhabdomyolysis, and circulatory collapse [167].

In light of all the evidence, it is clear that that caffeine consumption or supplementation can promote weight reduction; however, additional studies are necessary to explain all the mechanisms correlated with fat loss, improved BMI, and appetite reduction.

## 5. Conclusions

Obesity is rising all around the world. It is clear that fat accumulation is caused by several factors, such as genetic, epigenetic, and lifestyle factors, and the use of nutraceuticals for the treatment of this condition is constantly increasing. In fact, many products have been publicized and studied to investigate their effectiveness in the control and loss of body weight. Among these products, nutraceuticals enjoy great success among the population compared to anti-obesity drugs, in particular thanks to their low toxicity, competitive pricing, and lack of requirement for a medical prescription.

The mechanisms by which nutraceuticals can contribute to weight management are varied, including anti-inflammatory activities, inhibition of the sense of hunger, inhibition of fatty acid biosynthesis and/or intestinal fatty acid and cholesterol absorption levels, modulation of several pathways involved in carbohydrate digestion, fatty acid storage, insulin production and cellular glucose uptake, increase in energy expenditure, inhibition of adipocyte differentiation, lipolysis activation, and increasing of satiating effect. Through their potential to modify gene expression, certain nutraceuticals can assume the role of true epigenetic drugs, and their future potential must be explored through future clinical trials in humans. While it should not be forgotten that the effects of nutraceuticals are not such as to be able to mask a patient’s noncompliance with dietary and behavioral prescriptions, they can encourage adherence by accelerating achievement of the desired results.

## Figures and Tables

**Figure 1 plants-12-02273-f001:**
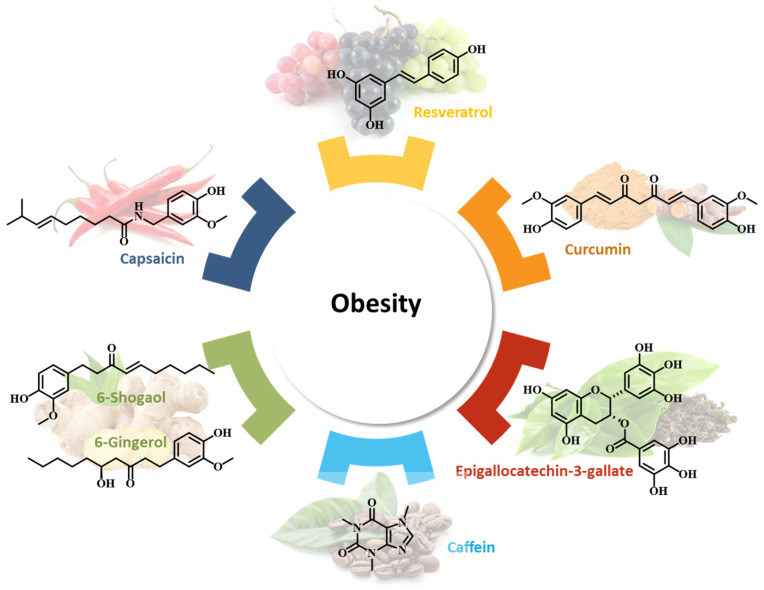
The main natural compounds involved in obesity management.

**Figure 2 plants-12-02273-f002:**
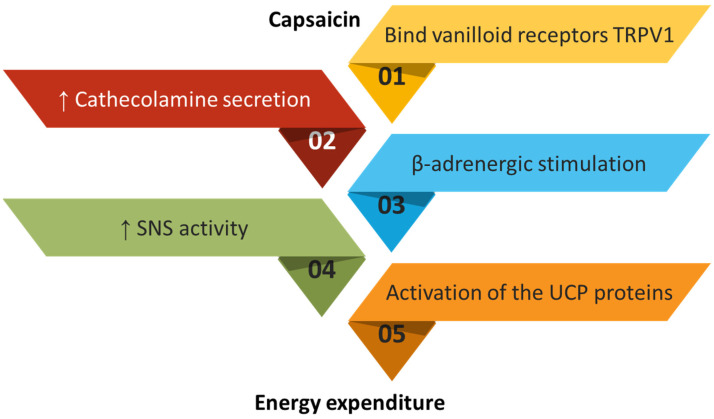
The mechanism of action of capsaicin. TRPV1 is excited by capsaicin, and the resulting signal is transmitted to the spinal cord through the afferent nerves it contains. The excitation of efferent nerves by the central nervous system causes an increased release of catecholamines (norepinephrine, epinephrine, and dopamine) at the adrenal level. In turn, catecholamines bind b-adrenergic receptors, increasing thermogenic activity. At gastrointestinal level, TRPV1 activation by capsaicin increases thermogenesis and activates UCP1 in BAT. Taking into account the exposed mechanisms, it is clear that capsaicin increases thermogenesis and energy expenditure via TRPV1 activation.

## Data Availability

Not applicable.

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
