# Peer review of "Plant-Derived Nutraceuticals Involved in Body Weight Control by Modulating Gene Expression"

_plants, 2023, doi:10.3390/plants12122273_

Round 1
Reviewer 1 Report
The review entitled “Plant derived nutraceuticals involved in body weight control by modulating gene expression” emphasizes the importance of nutraceuticals as a precise approach to weight management by regulating epigenetic changes. This approach will hand-pick nutraceuticals, strategize weight management policies and formulate interventions and therapeutics conferring to obesity.
The remarks are stated underneath:
a. Authors can include one more column in S1, about the effective doses or the recommended doses for adults.
b. Are there any studies that have mentioned about the toxic effects of these nutraceutical compounds if taken too much?
Author Response
Manuscript Number: plants-2420329
Manuscript title: Plant-derived nutraceuticals involved in body weight control by modulating gene expression
Authors: Maria Vrânceanu, Simona-Codruța Hegheș, Anamaria Cozma-Petruț, Roxana Banc, Carmina Mariana Stroia, Viorica Raischi, Doina Miere, Daniela-Saveta Popa, Lorena Filip
Response letter
Dear Reviewer #1,
We appreciate the useful comments and thoughtful suggestions provided. All have been carefully considered during revision.
A detailed response to your comments is outlined in the accompanying letter.
Best regards,
Assoc. Prof. Simona-Codruța Hegheș, Pharm., PhD
Reviewer #1 report:
The review entitled “Plant derived nutraceuticals involved in body weight control by modulating gene expression” emphasizes the importance of nutraceuticals as a precise approach to weight management by regulating epigenetic changes. This approach will hand-pick nutraceuticals, strategize weight management policies and formulate interventions and therapeutics conferring to obesity.
The remarks are stated underneath:
- Authors can include one more column in S1, about the effective doses or the recommended doses for adults.
Response: We thank the reviewer for the important suggestion. For the nutraceuticals addressed, we have added information concerning the effective, recommended and/or safe doses for adults. We opted for the introduction of this information in the manuscript and not in the Table S1, considering that in this manner we can provide even more complete and easily visible information about each nutraceutical.
- Are there any studies that have mentioned about the toxic effects of these nutraceutical compounds if taken too much?
Response: For the nutraceuticals addressed, we have performed a detailed literature research regarding the safety of their use. Where data related to safety issues, such as potential side effects, were identified, they were included in the revised manuscript.
Reviewer 2 Report
This manuscript entitled "Plant-derived nutraceuticals involved in body weight control by modulating gene expression" is very important because describes numerous nutraceuticals for weight loss. It is known that obesity, a metabolic syndrome associated with numerous co-morbidities, such as cardiovascular diseases, NAFLD, endocrine disorders, and others can be the main cause of death. In this review, the authors described in detail some nutraceuticals such as resveratrol, curcumin, epigallocatechin-3-gallate, ginger, capsaicin, and caffeine, which can alter gene expression, restoring the normal epigenetic profile and help weight loss.
Minor Comments:
For better monitoring of the role of epigenetic factors in the development of obesity, a scheme of pathogenetic mechanisms through which epigenetic factors contribute to obesity should be added. Also, in the same scheme or another, present all the mentioned nutraceuticals and mechanisms which modulate gene expression (mechanisms are given only for capsaicin.
Author Response
Manuscript Number: plants-2420329
Manuscript title: Plant-derived nutraceuticals involved in body weight control by modulating gene expression
Authors: Maria Vrânceanu, Simona-Codruța Hegheș, Anamaria Cozma-Petruț, Roxana Banc, Carmina Mariana Stroia, Viorica Raischi, Doina Miere, Daniela-Saveta Popa, Lorena Filip
Response letter
Dear Reviewer #2,
We appreciate the useful comments and thoughtful suggestions provided. All have been carefully considered during revision.
A detailed response to your comments is outlined in the accompanying letter.
Best regards,
Assoc. Prof. Simona-Codruța Hegheș, Pharm., PhD
Reviewer #2 report:
This manuscript entitled "Plant-derived nutraceuticals involved in body weight control by modulating gene expression" is very important because describes numerous nutraceuticals for weight loss. It is known that obesity, a metabolic syndrome associated with numerous co-morbidities, such as cardiovascular diseases, NAFLD, endocrine disorders, and others can be the main cause of death. In this review, the authors described in detail some nutraceuticals such as resveratrol, curcumin, epigallocatechin-3-gallate, ginger, capsaicin, and caffeine, which can alter gene expression, restoring the normal epigenetic profile and help weight loss.
Minor Comments:
For better monitoring of the role of epigenetic factors in the development of obesity, a scheme of pathogenetic mechanisms through which epigenetic factors contribute to obesity should be added. Also, in the same scheme or another, present all the mentioned nutraceuticals and mechanisms which modulate gene expression (mechanisms are given only for capsaicin).
Response: We thank the reviewer for the valuable suggestion. We developed the proposed scheme, in which we included the epigenetic processes that contribute to the pathogenesis of obesity. In the same scheme, we presented nutraceuticals involved in body weight control, along with mechanisms related to the modulation of gene expression through which these products can intervene in the weight loss process. Considering that the scheme is highly comprehensive and very representative of the content of our manuscript, we would like to present it as a graphical abstract.

Reviewer 3 Report
Authors should provide more epigenetic information in this manusript.
Authors should provide more epigenetic information in this manusript.
Author Response
Manuscript Number: plants-2420329
Manuscript title: Plant-derived nutraceuticals involved in body weight control by modulating gene expression
Authors: Maria Vrânceanu, Simona-Codruța Hegheș, Anamaria Cozma-Petruț, Roxana Banc, Carmina Mariana Stroia, Viorica Raischi, Doina Miere, Daniela-Saveta Popa, Lorena Filip
Response letter
Dear Reviewer #3,
We appreciate the useful comments and thoughtful suggestions provided. All have been carefully considered during revision.
A detailed response to your comments is outlined in the accompanying letter.
Best regards,
Assoc. Prof. Simona-Codruța Hegheș, Pharm., PhD
Reviewer #3 report:
Authors should provide more epigenetic information in this manuscript.
Response: We thank the reviewer for the thoughtful suggestion. We further deepened the topic of our review and added additional epigenetic information in the manuscript.
